# Fetal and neonatal outcomes in syphilis infected pregnant women in Reunion Island: An observational retrospective multicentric study

**Camille Cramez**, **Marine Lafont***, **Brahim Boumahni, Malik Boukerrou, Phuong Lien Tran***

Department of Gynecology and Obstetrics, University Hospital of La Réunion, Saint Pierre, Reunion Island, France

ʘ These authors contributed equally to this work.

* marine.lafont@chu-reunion.fr (ML); phuong.tran@chu-reunion.fr (PLT)

## Abstract

### Objectives

To evaluate the fetal and neonatal morbidity and mortality in pregnant women with syphilis during their pregnancy in Reunion Island, population benefiting from early and well-conducted screening and treatment.

### Methods

This is a retrospective observational study conducted in the four Reunion hospital centers between 2017 and 2022. The included patients were all pregnant patients with a biologically proven syphilitic infection and having given birth in one of the four centers mentioned.

### Results

108 patients were included, with 113 fetuses and newborns. Fetal mortality rate was 2.7%, neonatal mortality rate was 1.8% and congenital syphilis rate was 7%. Despite 37% of patients with a vulnerable psycho-social context, 72% were screened early, 13% in the second trimester, 44% had received optimal treatment and 48% optimal syphilis monitoring. We found 43% of obstetric complications with 12% of IUGR and 4.6% of pre-eclampsia. 21% of newborns were preterm and 17% small for gestational age. There was a significant trend between psycho-social vulnerability and congenital syphilis.

### Conclusions

Fetal and neonatal morbidity and mortality is low in our Reunion Island population where screening and treatment of syphilis are early and well conducted, although a vulnerable group of patients remain who require specific antenatal care.

**Data Availability Statement:** ***AT ACCEPT: Please ask authors if they would like all SI data files published alongside their paper. If yes, then please ask them to provide English translations and

confirm data are all de-identified.*** All relevant data are within the paper and its Supporting information files.

**Funding:** The author(s) received no specific funding for this work.

**Competing interests:** The authors have declared that no competing interests exist.

## Introduction

Syphilis has been on the rise again since the 2000s and, according to the World Health Organization (WHO), affects more than 7 million cases worldwide every year, with 425 cases of congenital syphilis per 100,000 live births [1]. According to the Lancet Stillbirth Series, syphilis is one of the leading preventable causes of in utero fetal death (IUFD), accounting for 7.7% of stillbirths worldwide [2].

The risk of maternal-fetal transmission of the disease begins at 14 weeks of gestation (WG). The more recent the maternal infection is, the greater the risk is. Nevertheless, the severity of fetal damage decreases with gestational age [3, 4]. Left untreated, syphilis is responsible for a high rate of fetal and neonatal complications: 40% IUFD, 20% neonatal death, 25% prematurity and intra-uterine growth retardation (IUGR), 20% post-natal sequelae [5–7]. The risk of obstetrical outcomes is also increased by up to 50%, with an OR greater than 2 in the case of late diagnosis and treatment [8, 9].

In a historical meta-analysis including six articles between 1917 and 2000, which studied the fetal and neonatal outcomes of syphilis in untreated syphilis patients compared with syphilis-free patients, syphilis patients were found to have 21% more IUFD, 9.3% more neonatal deaths, 10% more deaths in the first year of life and 50% more obstetric complications. In this study, carried out in Africa, the USA and the UK, more than half of all pregnancies with untreated syphilis were complicated [10].

The reference treatment is based on injectable BPG 2.4 million units (Extencillin®). Ideally started before 16 WG and at least four weeks before delivery, it reduces the risk of perinatal mortality or sequelae of congenital syphilis by more than 90%, enables ultrasound signs to regress, and prevents obstetrical complications by reducing the risk of vertical contamination [5, 11, 12].

In France, serological screening for syphilis is mandatory before 10 WG [13]. Since 2017, French authority and WHO have recommended repeat screening in the second trimester of pregnancy in the face of certain risk factors (multiple partners, history of sexually transmitted infections STIs, geographical areas with a high prevalence of infections low socio-economic level, absence of pregnancy follow-up) [14, 15].

In 2017, the prevalence of syphilis in pregnant women on Reunion Island was estimated at 0.11% [16]. The cumulative incidence of congenital syphilis is estimated to exceed 50 cases/100,000 live births [3].

Although recent literature has already highlighted the high risk of fetal and neonatal consequences of congenital syphilis in the absence of organized screening and effective treatment; to our knowledge, the risk of fetal and neonatal complications in patients with syphilis during pregnancy in a population benefiting from organized screening according to current recommendations in the first and second trimesters of pregnancy has never been studied.

The aim of our study is to assess the fetal and neonatal morbidity and mortality of women with syphilis during pregnancy in Reunion Island, a population that benefits from extensive and systematic screening, which is well conducted in the first and second trimesters of pregnancy.

## Material and methods

We conducted a multicenter retrospective observational study in the four hospital centers of Reunion Island (Centre Hospitalier Universitaire Nord et Sud—CHU, Centre Hospitalier Ouest Réunion—CHOR, Groupe Hospitalier Est Réunion—GHER), between January 1, 2017 and December 31, 2022, on fetal and neonatal morbidity and mortality in women with syphilis during pregnancy. Data were accessed for research purposes on May, 1 2023.

During consultation, verbal consent was obtained from patients to transcript their history in their medical records. Then, for this retrospective study of medical records, all data were fully anonymized before we accessed them. Authors had not access to information that could identify individual participants during or after data collection. This work was approved by the Research Ethics Committee of the University Hospital of Bordeaux, under the reference CER-BDX 2023–138.

## Study population

We included any pregnant patient with biologically proven syphilis infection who had given birth at one of the four above-mentioned centers. Serology was considered positive when TPHA was positive as well as VDRL titer ($\geq$ ½). Any pregnancy with missing neonatal data was excluded.

Patients were recruited via the files presented to the Multidisciplinary Center for Prenatal Diagnosis (CPDPN in France) and via a search by the Department of Medical Information (DIM) using the key words "pregnancy; syphilis". Under the regional protocol, all pregnant patients with syphilis are presented to the Indian Ocean CPDPN, guaranteeing exhaustive case registration.

## Data collection

Data were collected retrospectively from paper or computerized medical records at each hospital. All data are fully available without restriction (S1 and S2 Tables).

In order to meet our primary objective, our composite primary endpoint included fetal mortality, defined as spontaneous cessation of cardiac activity in utero at 14WG and above, and neonatal mortality, defined as death of a newborn at less than 28 days of age [17, 18].

Secondary objectives were multiple and consisted in assessing fetal and neonatal morbidity.

Antenatally, these were ultrasound signs of infectious fetopathy: oligo-anamnios, fetal anemia (hydramnios, ascites, pericardial effusion, hydrops, middle cerebral artery systolic peak > 1.5 MoM), bone mineralization disorders, hepatosplenomegaly, intestinal hyper-echogenicity, calcifications, cerebral malformations and IUGR. IUGR is defined as an estimated fetal weight below the 10th percentile (small for gestational age—SGA) associated with arguments in favor of a pathological growth defect [19].

For newborns, the secondary endpoints were:

- prematurity, defined as birth between 22 and 37WG of a child weighing over 500g [20];

- hypotrophy, defined by a birth weight < 10th percentile according to the Reunion Island perinatal network curve [21];

- microcephaly, defined by a cranial perimeter (CP) at birth < 3 standard deviations on the AUDIPOG curves [22, 23];

- the existence of congenital syphilis proven by biological and imaging examinations according to the Center for Diseases Control and Prevention (CDC) classification [11].

The CDC scenario classification, updated by the French National Reference Center (CNR), is described as follows:

- CDC scenario 1 "congenital syphilis confirmed or highly probable": newborn with clinical signs of congenital syphilis, positive PCR on biological sample (placenta or cord blood, nasal or oral secretions, skin lesion), positive IgM or VDRL four times higher than maternal VDRL.

- CDC scenario 2 "probable congenital syphilis": asymptomatic newborn whose VDRL is less than four times the maternal VDRL, and whose mother has been inadequately treated (treatment other than that of reference, treatment carried out < 4 weeks before delivery, undocumented, no serological decline or no treatment).

- CDC scenario 3 "possible congenital syphilis": asymptomatic newborn whose VDRL is less than four times the maternal VDRL and whose mother was correctly treated during pregnancy but after 16 months' gestation, with no argument for reinfection and no rise in VDRL titers.

- CDC scenario 4 "congenital syphilis unlikely": a newborn as described in scenario 3, but whose mother was properly treated before 16WG, with stable, low VDRL follow-up.

We collected maternal characteristics such as age, geographical origin, level of education, profession, marital status, psycho-social context, social security coverage, body mass index (BMI), gestity, parity, drug use, history of STI (gonorrhea, chlamydia, herpes, syphilis). A woman was identified with a vulnerable psycho-social background when she met any of the following criteria: domestic violence, incarcerated partner or partner with poly-intoxication, history of sexual assault or multiple abortions, psychological disorders, children in care or out of school, medico-psycho-social hospital file or housing difficulties.

We also collected history of obstetric complications such as prematurity, IUGR, IUF or pre-eclampsia. Pre-eclampsia is defined as systolic ($\geq$ 140 mmHg) and/or diastolic ($\geq$ 90 mmHg) gravid hypertension associated with significant proteinuria ($>$ 0.3g/24h) from 20WG onwards [24].

In addition, we collected data concerning the pregnancy such as dating, co-infections, obstetrical complications as described above and the context of syphilis screening (place, reason, term, first VDRL, stage). Screening was considered successful if it was carried out in the first trimester and then repeated in the second trimester of pregnancy. Treatment modalities were recorded (number and timing of injections, adverse effects, prevention of Herxeimer reaction according to recommendations). Treatment was considered optimal if the treatment was appropriate for stage of syphilis, given before 16 WG and more than 30 days before delivery. Follow-up of syphilis during pregnancy was considered optimal if the patient was properly addressed to infectious disease (CEGGID) and Antenatal Diagnosis (DAN) departments, with monthly ultrasound and VDRL, but also if psychosocial care was provided, according to the regional protocol [25].

Partner screening and treatment were specified. Treatment of the newborn was also recorded.

## Statistical analysis

The data, anonymized by identification number and classified using Excel spreadsheet, were analyzed using P-value and XLstat softwares. They are presented as numbers and proportions (%) for categorical variables, and as means and standard deviations for continuous variables for descriptive data analysis.

## Results

Between January 1, 2017 and December 31, 2022, we included 124 patients in the four hospital centers in Reunion Island. Sixteen patients whose pregnancy outcome was unknown were excluded. Our final cohort therefore comprised 108 patients.

## Maternal characteristics

The majority of patients were from Reunion Island (n = 92, 85.6%), and most lived in the south of the island (n = 52, 48%). The mean age was 24.2 years old, with a minimum age of 14 years old and a maximum age of 44 years old. The majority had a low socio-economic status, and 37% (n = 39) had a vulnerable psycho-social background. None of the patients declared that they had protected intercourse, with 24% of patients having a history of STI; in particular, 11% of patients had a history of syphilis, 75% of them in the year preceding pregnancy, but all had been treated (Table 1).

**Table 1. Characteristics of the study population.**

|  | *Study population*<br>*N = 108 / (%)* |
|---|---|
| **Age (years)** |  |
| Mean ± standard deviation | 24.2 ± 6.38 |
| < 20 years | 25 (23) |
| 20–25 years | 53 (49) |
| 26–30 years | 12 (11) |
| > 30 years | 18 (16) |
| **Place of birth** |  |
| Reunion Island | 92 (85.6) |
| France | 4 (3.7) |
| Madagascar | 5 (4.6) |
| Mayotte | 5 (4.6) |
| Comores | 1 (0.93) |
| Mauritius | 1 (0.93) |
| **Place of Residence on Reunion Island** |  |
| South | 52 (48) |
| North | 26 (24) |
| West | 13 (12) |
| East | 16 (15) |
| **Mayotte** | 1 (0.93) |
| **Education level** |  |
| High school | 80 (74) |
| Bachelor | 22 (20) |
| College | 6 (5.6) |
| **Occupation** |  |
| Without | 92 (85) |
| Employee | 7 (6.5) |
| Student | 8 (7.4) |
| Senior manager | 1 (0.93) |
| **Social coverage** |  |
| Without | 7 (6.5) |
| Solidarity regime | 76 (70) |
| Social security | 19 (18) |
| Social security + mutual company | 6 (5.6) |
| **Marital status** |  |
| Married | 13 (12) |
| Cohabitation | 45 (42) |
| Single | 48 (44) |

*(Continued)*

**Table 1.** (Continued)

|  | *Study population*<br>*N = 108 / (%)* |
|---|---|
| Divorced | 2 (1.9) |
| **Psycho-social background** |  |
| Total | 39 (37) |
| At least two associated factors | 14 (13) |
| Spousal or domestic violence | 14 (13) |
| Incarcerated spouse or legal proceedings | 6 (5.7) |
| Spouse with poly-intoxication | 7 (6.5) |
| History of sexual assault | 4 (3.7) |
| History of multiple abortions | 4 (3.7) |
| Psychological vulnerability | 3 (2.7) |
| Children in care or out of school | 7 (6.5) |
| Medico-psycho-social file | 14 (13) |
| Housing difficulties | 6 (5.7) |
| **Use of toxic substances** |  |
| Total | 27 (25) |
| Tobacco | 19 (18) |
| Alcohol | 2 (1.9) |
| Cannabis | 6 (5.7) |
| Poly-consumption | 5 (4.6) |
| Multiple partners | 8 (7.4) |
| **History of STIs** |  |
| Total | 26 (24) |
| Syphilis | 12 (11) |
| *Treated less than 1 year ago* | 8/12 (75) |
| *Treated more than 1 year ago* | 4/12 (25) |
| *Chlamydiae* | 9 (8.3) |
| *Gonorrhea* | 4 (3.4) |
| *Herpes* | 1 (0.93) |
| **BMI** |  |
| *Mean ± standard deviation* | 26.1 ± 6.81 |
| **Gestity** |  |
| *Mean ± standard deviation* | 2.79 ± 2.08 |
| **Parity** |  |
| *Mean ± standard deviation* | 1.14 ± 1.54 |
| **History of obstetrical complications** |  |
| Total | 17 (16) |
| Pre-eclampsia | 1 (0.93) |
| Prematurity | 8 (7.4) |
| IUGR | 6 (5.6) |
| IUFD | 2 (1.9) |

## Pregnancy description

85 patients (79%) had a normal pregnancy follow-up, even though 13 of them were dated late (12%). Nevertheless, 17% of patients had irregular pregnancy follow-up and less than 2% had no follow-up at all (Table 2).

**Table 2. Characteristics of the pregnancy.**

|  | *Number of women*<br>*N = 108 / (%)* |
|---|---|
| Late dating | 24 (22) |
| **Start of pregnancy follow-up** |  |
| First trimester | 83 (77) |
| Second trimester | 22 (20) |
| Third trimester | 1 (0.93) |
| No follow-up | 2 (1.9) |
| **Pregnancy follow-up** |  |
| Regular | 72 (67) |
| Regular after late dating | 13 (12) |
| Irregular | 18 (17) |
| Complete discontinuation | 5 (4.6) |
| **Use of toxic substances** |  |
| Total | 25 (23.6) |
| Tobacco | 20 (19) |
| Alcohol | 2 (1.9) |
| Cannabis | 3 (2.7) |
| Poly-consumption | 5 (4.6) |

Initial screening was generally performed during the first trimester of pregnancy, but in 13% of patients, it was only performed during the 6th month of pregnancy. The average term of the first syphilis management appointment was around 16.8 WG (σ = 9.35WG), mostly at the CEGGID (n = 48, 44%) and then at the maternity hospital (n = 35; 32%). Syphilis was mainly latent, although 9.2% were active (Table 3).

Syphilis treatment was considered optimal in 44% of patients, with 55% of injections given before 16WG, 88% more than 30 days before delivery, and 88% the correct number of injections according to stage. Syphilis surveillance was considered suboptimal in 52% of patients (Table 4).

## Fetal mortality and morbidity

Fetal mortality was 2.7% (n = 3). Interestingly, in the only case where the ultrasound was regular and treatment optimal, IUFD occurred at 18WG. In the second case of IUFD, which occurred at 25WG, the fetus, having received non-optimal treatment, presented intestinal hyper-echogenicity associated with ventriculomegaly and IUGR. The third patient concerned was from Mayotte and had also been inadequately treated. The fetus, from a twin pregnancy, died in utero at 29WG, in a state of hydrops. Amniocentesis revealed a positive T. Pallidum PCR.

In terms of fetal morbidity, 31% of pregnancies were marked by ultrasound anomalies suggestive of infectious fetopathy, with 13% showing fetal anemia and 19% IUGR. 12.9% of fetuses had at least two associated ultrasound anomalies.

In terms of obstetric complications, 4.6% of pregnancies were complicated by pre-eclampsia. (Table 5, Fig 1).

## Neonatal mortality and morbidity

We included 113 newborns (with 10 from 5 twin pregnancies), with an estimated neonatal mortality rate of 1.7% (n = 2), but no newborn died between 7 and 28 days of age. Of these two

**Table 3. Syphilis diagnoses.**

| | Number of women N = 108 / % |
|---|---|
| **Reason for first screening** | |
| Discovery of pregnancy | 15 (14) |
| STI screening | 5 (4.6) |
| Systematic first trimester screening | 62 (57) |
| Second trimester screening | 14 (13) |
| Clinical signs of syphilis | 2 (1.9) |
| Syphilis in the partner | 3 (2.7) |
| Ultrasound signs suggestive of syphilis | 2 (1.9) |
| Delivery | 5 (4.8) |
| **Place of first screening** | |
| Maternity hospital | 35 (32) |
| CEGGID | 48 (44) |
| General practitioner | 17 (16) |
| Private midwife | 4 (3.7) |
| Private gynecologist | 4 (3.7) |
| **Term of pregnancy of first consultation (WG)** | |
| *Mean ± standard deviation* | 16.8 ± 9.35 |
| **Staging** | |
| Early latent | 38 (35) |
| Primary | 8 (7.4) |
| Secondary | 2 (1.8) |
| Late latent | 60 (56) |
| **First VDRL** | |
| *Median [Q25-75]* | 8 [4; 16] |
| **Co-infection** | |
| Chlamydia | 19 (18) |
| Gonorrhea | 2 (1.8) |
| Mycoplasma | 8 (7.4) |
| Trichomonas | 2 (1.8) |
| Herpes | 1 (0.93) |
| Scabies | 1 (0.93) |
| > 2 co-infections | 9 (8.3) |
| **Partner** | |
| Screening | 81 (75) |
| Treatment | 79 (73) |

neonatal deaths, classified as CDC scenario 1, and whose pregnancies had been marked by sub-optimal treatment, one newborn died at one hour of life in a context of IUGR with hydrocephalus associated with extreme prematurity at 25WG. The second died at three days of age after a premature birth at 32WG in a context of hydrops with multivisceral failure.

With regard to neonatal morbidity, the mean term of birth was 37.6WG, with 21% premature births. Fifteen newborns were managed at birth by pediatricians for respiratory distress, and two had neurological signs labeled "fetal distress" or "neurosyphilis". There were no other clinical signs suggestive of congenital syphilis at birth.

Eight newborns had a diagnosis of congenital syphilis (7%), i.e. a CDC 1 scenario assigned by practitioners. Among the 44% of patients who were well treated, we found two cases of

**Table 4. Syphilis follow-up.**

| | Number of women N = 108 / % |
|---|---|
| **Treatment** | |
| Optimal treatment | 47 (44) |
| Treatment before 16 WG | 60 (55) |
| Adequate number of injections | 95 (88) |
| Average number of injections ± *standard derivation* | 2.66 (0.726) |
| Average term of injections ± *standard derivation* | 17.8 (9.44) |
| Undesirable effects | 1 (0.93) |
| Herxeimer reaction prevention | 19 (18) |
| **Treatment—delivery time** | |
| < 30 days | 13 (12) |
| **Follow-up** | |
| **Optimal follow-up** | 52 (48) |
| Antenatal diagnosis follow-up (DAN) | 97 (90) |
| Average number of ultrasound exam ± *standard derivation* | 5.56 ± 2.82 |
| Average number of CEGGID appointments ± *standard derivation* | 2.19 ± 1.64 |
| Monthly VDRL testing | 68 (63) |
| Psycho-social follow-up | 40 (37) |
| Psychosocial follow-up proposed but not desired by the patient | 46 (43) |
| **Sub-optimal follow-up** | 56 (52) |
| Patient failure | 14 (13) |
| Caregiver failure | 35 (32) |
| Failure of caregiver and patient | 7 (6.5) |
| **Birth tests** | |
| None | 35 (33) |
| VDRL | 35 (33) |
| > 2 exams (IgM, PCR, fundus, X-rays, ultrasound) | 42 (40) |
| **Newborn VDRL at birth if assayed** | |
| *Mean ± sandard deviation* | 6.62 (36.9) |
| **Neonatal treatment** | |
| Extencilline IV treatment | 25 (22) |
| Average treatment time ± *standard derivation* | 1.19 ± 3.605 |

congenital syphilis, including one IUFD; while among the 56% of patients who were inadequately treated, we found six cases of congenital syphilis (Table 6, Fig 2).

The five deceased fetuses and newborns were from mothers with vulnerable psycho-social backgrounds. For two of them, syphilis had been diagnosed during the systematic first-trimester screening; for the other two, screening had been carried out in the presence of suspicious echographic signs, and for the last fetus, serology had been carried out at delivery.

## Discussion

The prevalence of congenital syphilis in our study was 7%, while it was 15% in the cohort of pregnant women with untreated syphilis by Gomez et al. [10]. Comparison with congenital syphilis rates in other studies in Reunion is difficult, as only one of them used the recommended CDC classification, updated by the CNR. This study had classified 18% of newborns in CDC scenario 1, while two other studies had classified 7% and 19.6% of newborns in

**Table 5. Fetal mortality and morbidity.**

| | Number of fetuses N = 113 / % |
|---|---|
| **Fetal mortality rate** | 3 (2.7) |
| **Ultrasound abnormalities** | |
| Abnormalities suggestive of infectious fetopathy | 33 (31) |
| Oligo-anamnios | 5 (4.6) |
| Signs of fetal anemia | 14 (13) |
| Hepatosplenomegaly | 3 (2.7) |
| Intestinal hyperechogenicity, calcifications | 4 (3.7) |
| Bone mineralization disorders | 1 (0.93) |
| IUGR | 21 (19) |
| Cerebral malformations | 3 (2.7) |
| Macrosomia | 4 (3.7) |
| > 2 associated anomalies | 14 (12.9) |
| **Obstetrical complications** | 46 (43) |
| Pre-eclampsia | 5 (4.6) |
| IUGR | 13 (12) |
| > 2 associated complications | 11 (10) |

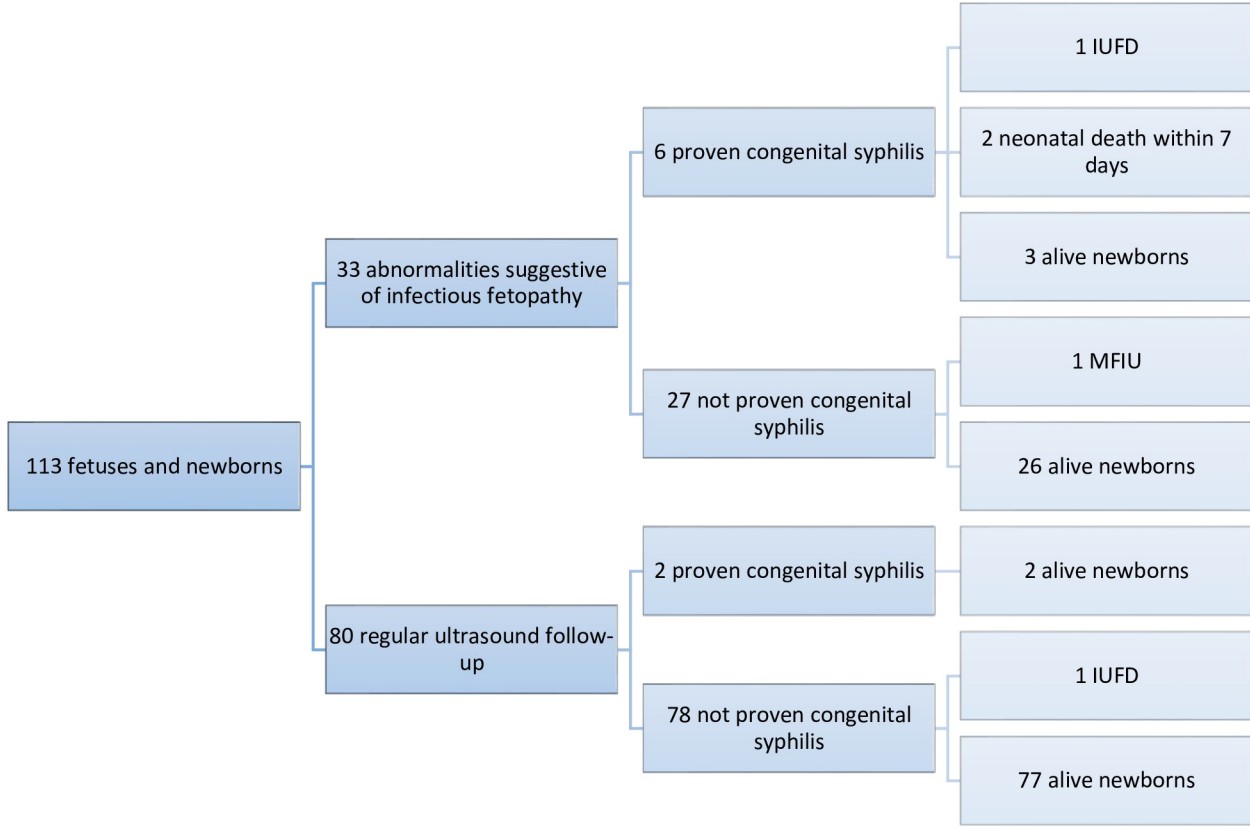

**Fig 1. Fetal and neonatal issue according to ultrasound findings.**

**Table 6. Neonatal mortality and morbidity.**

|  | Number of newborns N = 113 / % |
|---|---|
| **Neonatal mortality rate** |  |
| Death < 7 days | 2 (1.7) |
| Death between 7 and 28 days | 0 (0) |
| **Birth term (WG)** |  |
| *Mean ± standard deviation* | 37.6 (3.86) |
| Preterm birth < 37WG | 24 (21) |
| **Sex** |  |
| Female | 59 (52) |
| Male | 54 (48) |
| **Birth weight (g)** |  |
| Mean ± standard deviation | 2822 (703) |
| Low birth weight | 19 (17) |
| **Birth CP (cm)** |  |
| Mean ± standard deviation | 33 (2.44) |
| Microcephaly | 10 (8.9) |
| **Birth size (cm)** |  |
| Mean ± standard deviation | 47 (4.26) |
| **Birth clinical signs** |  |
| Respiratory distress | 15 (14) |
| Neurological abnormalities | 2 (1.9) |
| **CDC Scenario** |  |
| 1 "confirmed or very probable congenital syphilis" | 8 (7) |
| 2 "probable congenital syphilis" | 11 (9.7) |
| 3 "possible congenital syphilis" | 21 (19) |
| 4 "congenital syphilis unlikely" | 73 (65) |

2013 and 2018 respectively as "probable congenital syphilis" [16, 26, 27]. Nevertheless, the prevalence of congenital syphilis in our study matches with recent results founded abroad [28, 29].

Our results show a lower fetal mortality rate than previous reported in Reunion literature. A study of 85 pregnant syphilis patients found a fetal mortality rate of 7%, compared with 2.7% in our study. However, carried out just after the syphilis upsurge of the 2000s, this study included almost twice as many patients with an unfavorable socio-economic background as our study (68% vs. 37%). In addition, there was a higher proportion of active syphilis (24% vs. 9.2%) [27]. A second monocentric study carried out between 2008 and 2014 in the same type of population found a fetal mortality rate of 5%, with 74% of the patients included receiving inappropriate treatment, compared with 56% in our study [26]. Over the last twenty years, the systematic reporting of syphilis cases during pregnancy to the CPDPN and the optimization of treatment seems to have reduced the fetal mortality rate in Reunion Island.

Fetal mortality rates in developing countries are expected to be high. Indeed, a Guyanese study carried out between 1992 and 2004 on 85 patients from very vulnerable socio-economic backgrounds with syphilis during pregnancy, 40% of whom were poorly monitored, with late detection and treatment, showed a fetal mortality rate of 12.9% [30]. This was also the case in a Chinese study carried out between 2011 and 2018 on 3,474 pregnant syphilis patients, which found a fetal mortality rate of 3.8%. However, the authors point out that 16% of patients had

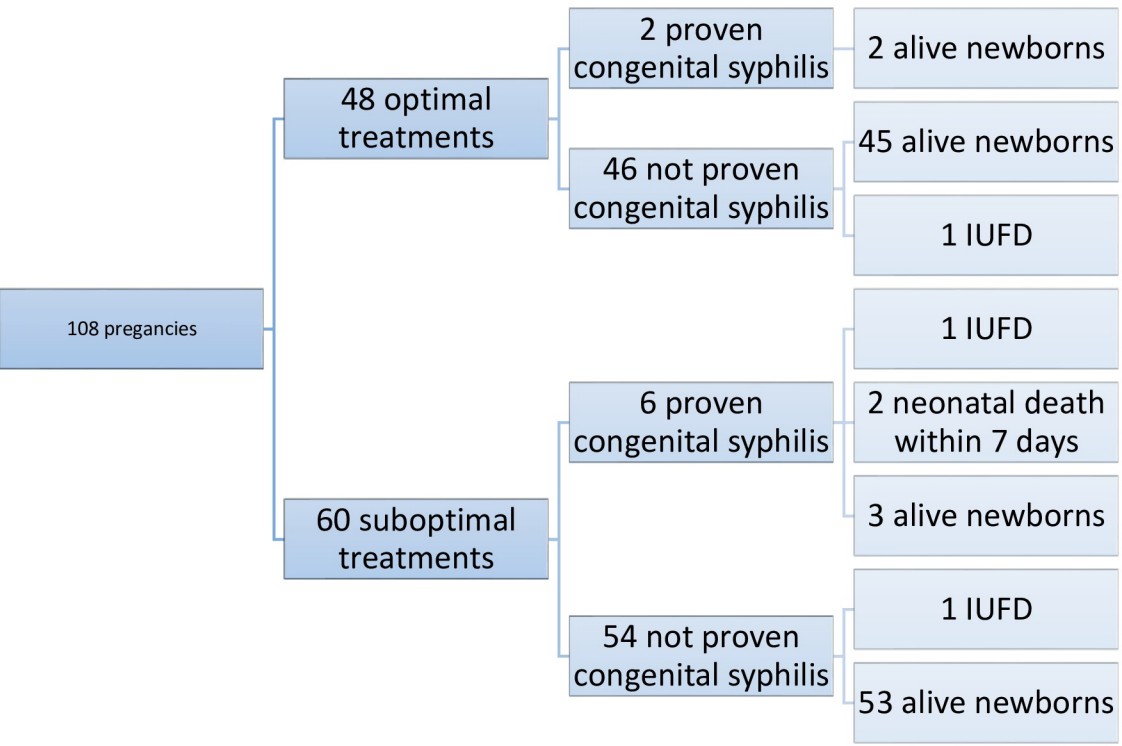

**Fig 2. Fetal and neonatal issue according to treatment.**

not been treated, and more than half had not been treated according to WHO recommendations [31].

Few articles assess the mortality of newborns from mothers with syphilis. Surprisingly, the neonatal mortality rate in our study is higher than in recent literature. First, in Reunion Island, the previously cited authors reported no neonatal deaths. Second, at international level, three recent studies are available. A first Chinese study found an early neonatal death rate of 0.8%, compared with 1.7% in our study. A second Chinese study reported only one case of neonatal death among 682 newborns included, as did a third Italian study, which found only one case of neonatal death among 323 newborns included [28, 29, 32].

However, our fetal and neonatal mortality rates are low compared with historical data, notably the meta-analysis by Gomez et al, which found fetal mortality rates between 17 and 44%, and neonatal mortality rates between 6.6 and 13.6%, among patients with syphilis during pregnancy, without any treatment [10].

Concerning fetal morbidity, we found the same rate of ultrasound anomalies suggestive of infectious fetopathy during follow-up as in the Reunion study by Baranzelli et al (31%) [27]. Nevertheless, the Italian study of Salomé et al found a suprising ultrasound abnormality rate of 5% among twenty cases of proven congenital syphilis, reinforcing the fact that the absence of ultrasound signs does not rule out congenital syphilis [28].

We found the same obstetrical complication rates than Le Chevalier de Preville et al study in Reunion Island (43% vs. 45%). These rates are significantly lower than the historical rates found in untreated syphilitic pregnant patients (54–82%) [10, 26].

In terms of neonatal morbidity, the prematurity rate for newborns born to mothers not treated for syphilis can be as high as 25% to 30% [5, 10]. Our prematurity rate is slightly

lower (21%), but in line with recent studies in Reunion (22–28%) [26, 27]. However, these prematurity rates are higher than those found in French, Reunion and international perinatal surveys [33]. Our results suggest that this high rate of prematurity is not only linked to syphilis, but also to exposure to known risk factors for prematurity (young maternal age, single status, low educational and socio-economic level, lack of access to antenatal care). Indeed, there appears to be a significant trend between vulnerable psycho-social background and congenital syphilis [31].

In addition, a Brazilian study from 2020 investigating the risk factors for prematurity specifically in cases of congenital syphilis, also described a young, psycho-socially fragile maternal population. The authors noted a prematurity rate of 15.3%, among mothers a quarter of whom had no follow-up at all, a quarter of whom had started follow-up late and 60% of whom had not been screened for syphilis. Drug use, lack of prenatal follow-up and a fortiori of screening and treatment, positive serology and, above all, a VDRL > 1/8 at delivery (reflecting an early and high bacteremia), were identified as significant risk factors for prematurity in congenital syphilis [34].

Finally, the vulnerable socio-economic background of mothers with syphilis, as a risk factor for prematurity, may also be associated with low birth weight. This was the case in a Chinese study which found 57.4% of hypotrophic newborns in syphilis-infected patients, treated in only 7.8% of cases [29]. In Reunion Island, this rate of neonatal hypotrophy is lower, and stable since 2014 (18% versus 17% in our study) [26].

## Conclusion

Our study found low fetal and neonatal mortality rates among pregnant syphilis patients, with a congenital syphilis prevalence of 7%. This suggests the effectiveness of a well-managed health policy against syphilis, including early detection and treatment.

According to our results, fetal and neonatal morbidity due to syphilis is low in Reunion Island; and syphilis screening, treatment and follow-up appears to be optimal in comparison with previous literature. However, no study had been carried out to confirm the effectiveness of syphilis eradication strategies recommended by the WHO for pregnant syphilis patients. Our study supports these recommendations.

Nevertheless, a small part of the population, represented by young women with vulnerable psycho-social backgrounds, must always be carefully targeted to ensure exhaustive screening of the whole population and to protect mother and child health.

## Supporting information

**S1 Table. Anonymized data syphilis in Reunion Island.**
(XLSX)

**S2 Table. Statistical analyses syphilis in Reunion Island.**
(XLSX)

## Author Contributions

**Conceptualization:** Camille Cramez, Marine Lafont.

**Data curation:** Camille Cramez.

**Funding acquisition:** Phuong Lien Tran.

**Methodology:** Marine Lafont.

**Resources:** Brahim Boumahni.

**Supervision:** Marine Lafont.

**Validation:** Malik Boukerrou.

**Visualization:** Brahim Boumahni.

**Writing – original draft:** Camille Cramez.

**Writing – review & editing:** Marine Lafont, Phuong Lien Tran.

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
