## [Decision Letter · Decision Letter 0]

12 Apr 2024

PONE-D-24-05431Fetal and neonatal outcomes in syphilis infected pregnant women in Reunion Island : an observationnal retrospective multicentric study.PLOS ONE

Dear Dr. CRAMEZ,

Thank you for submitting your manuscript to PLOS ONE. After careful consideration, we feel that it has merit but does not fully meet PLOS ONE’s publication criteria as it currently stands. Therefore, we invite you to submit a revised version of the manuscript that addresses the points raised during the review process.

We look forward to receiving your revised manuscript.

Kind regards,

Everton Falcão de Oliveira, Ph.D

Academic Editor

PLOS ONE

Journal Requirements:

Reviewers' comments:

Reviewer's Responses to Questions

**Comments to the Author**

1. Is the manuscript technically sound, and do the data support the conclusions?

Reviewer #1: Yes

2. Has the statistical analysis been performed appropriately and rigorously? 

Reviewer #1: Yes

3. Have the authors made all data underlying the findings in their manuscript fully available?

Reviewer #1: Yes

4. Is the manuscript presented in an intelligible fashion and written in standard English?

Reviewer #1: Yes

5. Review Comments to the Author

Reviewer #1: I have had the opportunity to read your manuscript entitled “Fetal and neonatal outcomes in syphilis infected pregnant women in Reunion Island : an observationnal retrospective multicentric study.”, and I appreciate the effort you have put into this study and the reporting of the numerous variable pertinent to prenatal screening and obstetric care with linkage to the newborn. I feel the results presented in the paper would be of interest to readers and are a contemporary report of success in mitigating congenital syphilis with routine screening and management of this infection.

The methodology is appropriate for the study and the reporting of these findings is thorough. The conclusions drawn from the results are suitable. I believe that reporting these findings would be of value to the scientific community in helping others measure success of screening and treatment protocols for congenital syphilis.

However, I would like to suggest some revisions to improve the readability of the paper for English-speaking audiences. While the content is scientifically robust, the language could benefit from some editorial changes. These changes would not only enhance the clarity of your arguments but also ensure that your findings are effectively communicated to the reader.

Here are a few areas that could use some attention:

1. Grammar and Syntax: There are instances where the sentence structure could be improved for better readability. For example, the following sentences caught my eye.

144: No re-ascension would be better stated as “no rise in VDRL titers”

158: “the number of injections was adapted to the stage of syphilis” could be better stated as if the treatment was appropriate for stage of syphilis.

163: this reads poorly, I would work this sentence into the preceding paragraphs.

181-182: for an English audience, “protected their sexual relations” is also a little oddly put, “had protected intercourse”, or simply “used protection consistently”.

Other additional comments are as noted below:

General terms of vulnerable socio-ecomonic background are hard to interpret given that there appears to be no clear definition given in the paper.

88-89: authors report that there is systematic screening which is well conducted, and would like to see some numbers which relate to this assertion. In the US many states have mandated screening three times in pregnancy, and despite laws being in place there have been surges suggesting that screening may not be as good as they/we expect.

106: Excluding pregnancy with missing neonatal data may bias these results, curious if there is an approximation of how many pregnancies had missing neonatal data and why this would be the case. I am assuming that perhaps this means any follow up data at 28 days of age, which if so, might just be good to just state as such, because missing neonatal data could be weight, or length, or gestational age, etc..

180-181: While I can assume what one would mean by a “vulnerable psycho-social background” by reading the table, it might be good to define or state within the text.

197: looks like 16SA needs to be 16 WG (again this is also up at line 146)

Figure 1: I think I understand what the authors mean by with and without congenital syphilis, the problem with this is the probabilistic nature of the infection they note earlier in the manuscript. I would recommend “not proven” or something of the like. Alternatively could expand out to have all 4 scenarios.

Results: I am interested to see so many individuals with “congenital syphilis unlikely” as noted in table 6. If many of these women were receiving treatment during pregnancy, then per the CDC guideline, the children could not be classified as such as the guideline states “Scenario 4: Congenital Syphilis Unlikely

Any neonate who has a normal physical examination and a serum quantitative nontreponemal serologic titer equal to or less than fourfold of the maternal titer at delivery and both of the following are true:

The mother’s treatment was adequate before pregnancy. The mother’s nontreponemal serologic titer remained low and stable (i.e., serofast) before and during pregnancy and at delivery (e.g., VDRL ≤1:2 or RPR ≤1:4). I am totally open to the possibility that I am misinterpreting this but in general treatment during pregnancy precludes being classified as unlikely. I do not feel that this appreciably changes the results from the paper, but I do think it is worth mentioning. Furthermore, if the

I hope you find these suggestions helpful. I look forward to seeing the revised manuscript and believe that these changes will significantly enhance the impact of your work.

Best regards,

6. PLOS authors have the option to publish the peer review history of their article (what does this mean?). If published, this will include your full peer review and any attached files.

Reviewer #1: No

---

## [Author Response · Author response to Decision Letter 0]

13 Aug 2024

1) Reviewer: � Authors have no financial disclosure to declare.

2) Reviewer: We recommend that you deposit your laboratory protocols in protocols.io to enhance the reproducibility of your results.

The laboratory protocol has been deposited on protocols.io under and is successfully published, under the DOI : https://dx.doi.org/10.17504/protocols.io.6qpvr83mblmk/v1

3) Reviewer: PLOS ONE's style requirements, including those for file naming. The PLOS ONE style templates can be found at 

Manuscript has been revised according to PLOS ONE Style Requirements.

The first page, with authors affiliation, has been changed, with 2 e-mail addresses left (Phuong Lien TRAN and Camille CRAMEZ). The mention “MD” and “PhD” have been removed. We wanted to mention that Camille CRAMEZ, Marine LAFONT and Phuong Lien TRAN contributed equally to this work.

4) Reviewer: We note that you have indicated that there are restrictions to data sharing for this study. For studies involving human research participant data or other sensitive data, we encourage authors to share de-identified or anonymized data. However, when data cannot be publicly shared for ethical reasons, we allow authors to make their data sets available upon request. For information on unacceptable data access restrictions, please see http://journals.plos.org/plosone/s/data-availability#loc-unacceptable-data-access-restrictions. 

If there are ethical or legal restrictions on sharing a de-identified data set, please explain them in detail (e.g., data contain potentially identifying or sensitive patient information, data are owned by a third-party organization, etc.) and who has imposed them (e.g., a Research Ethics Committee or Institutional Review Board, etc.). Please also provide contact information for a data access committee, ethics committee, or other institutional body to which data requests may be sent.

If there are no restrictions, please upload the minimal anonymized data set necessary to replicate your study findings to a stable, public repository and provide us with the relevant URLs, DOIs, or accession numbers. Please see http://www.bmj.com/content/340/bmj.c181.long for guidelines on how to de-identify and prepare clinical data for publication. For a list of recommended repositories, please see https://journals.plos.org/plosone/s/recommended-repositories. You also have the option of uploading the data as Supporting Information files, but we would recommend depositing data directly to a data repository if possible.

“The PLOS Data policy requires authors to make all data underlying the findings described in their manuscript fully available without restriction, with rare exception (please refer to the Data Availability Statement in the manuscript PDF file). The data should be provided as part of the manuscript or its supporting information, or deposited to a public repository. For example, in addition to summary statistics, the data points behind means, medians and variance measures should be available. If there are restrictions on publicly sharing data—e.g. participant privacy or use of data from a third party—those must be specified. »

After discussion between the authors, it did not seem unreasonable to release the data, given that no institution has forbidden it, subject to correct anonymization. Please find attached the Excel document with anonymized data collection and statistical tables.

5) Reviewer: Please amend either the title on the online submission form (via Edit Submission) or the title in the manuscript so that they are identical 

It is done, the two titles are identical, sorry for the oversight.

6) Reviewer: Please review your reference list to ensure that it is complete and correct. If you have cited papers that have been retracted, please include the rationale for doing so in the manuscript text, or remove these references and replace them with relevant current references. Any changes to the reference list should be mentioned in the rebuttal letter that accompanies your revised manuscript. If you need to cite a retracted article, indicate the article’s retracted status in the References list and also include a citation and full reference for the retraction notice.

All reference have been reviewed and can be found on the web. They all are recent articles. If you do not find one of them, may I ask you to tell me, so that I could help you?

7) Reviewer: modifications in more literary English

144: No re-ascension would be better stated as “no rise in VDRL titers”

158: “the number of injections was adapted to the stage of syphilis” could be better stated as if the treatment was appropriate for stage of syphilis.

181-182: for an English audience, “protected their sexual relations” is also a little oddly put, “had protected intercourse”, or simply “used protection consistently”.

197: looks like 16SA needs to be 16 WG (again this is also up at line 146)

Figure 1: I think I understand what the authors mean by with and without congenital syphilis, the problem with this is the probabilistic nature of the infection they note earlier in the manuscript. I would recommend “not proven” or something of the like. Alternatively could expand out to have all 4 scenarios.

Changes have been made.

163: this reads poorly, I would work this sentence into the preceding paragraphs

The sentence has been modified: “Follow-up of syphilis during pregnancy was considered optimal if the patient was properly addressed to infectious disease (CEGGID) and Antenatal Diagnosis (DAN) departments, with monthly ultrasound and VDRL, but also if psychosocial care was provided, according to the regional protocol.”

General terms of vulnerable socio-ecomonic background are hard to interpret given that there appears to be no clear definition given in the paper.

180-181: While I can assume what one would mean by a “vulnerable psycho-social background” by reading the table, it might be good to define or state within the text.

A definition for the term “vulnerable socio-economic background” has been added : “A woman was identified with a vulnerable psycho-social background when she met any of the following criteria: domestic violence, incarcerated partner or partner with poly-intoxication, history of sexual assault or multiple abortions, psychological disorders, children in care or out of school, medico-psycho-social hospital file or housing difficulties.”

8) Reviewer: 88-89: authors report that there is systematic screening which is well conducted, and would like to see some numbers which relate to this assertion. In the US many states have mandated screening three times in pregnancy, and despite laws being in place there have been surges suggesting that screening may not be as good as they/we expect.

The 2021 La Réunion perinatal survey revealed that almost 99% of pregnant women were screened for syphilis. You may see on table 19 below that between 98 and 99.9% of pregnant women have been screened for syphilis in La Réunion. You may find the percentage for mainland France just at the right of these numbers.

https://www.santepubliquefrance.fr/determinants-de-sante/sante-sexuelle/documents/enquetes-etudes/sante-perinatale-a-la-reunion.-resultats-de-l-enquete-nationale-perinatale-2021-dans-les-drom-enp-drom-2021

9) Reviewer: « Excluding pregnancy with missing neonatal data may bias these results, curious if there is an approximation of how many pregnancies had missing neonatal data and why this would be the case. I am assuming that perhaps this means any follow up data at 28 days of age, which if so, might just be good to just state as such, because missing neonatal data could be weight, or length, or gestational age, etc..”

Unfortunately, it is true that it may bias our results.

On 124 patients screened by the Department of Medical Information (DIM) and in the CPDPN files, 11 must have been excluded because medical files were totally incomplete. In La Réunion, medical files are informatized and it was not acceptable that the information was not completely filled in.

Moreover, 5 patients intended to be included have to be excluded because neonatal data missed such as, indeed, birth weight, birth length, birth cranial perimeter, neonatal exams to determine the CDC scenario. It can be explained by two factors in addition of an uncompleted file by human error. The first one is the lack of experience in case of suspicion of congenital syphilis in small centers, outside the University Hospital. The second one is that when newborns are taken into intensive care, all the biometrics are not measured in delivery room and then theses information lack in informatized files.

I hope this sounds clear to you.

10) Revi ewer: I am interested to see so many individuals with “congenital syphilis unlikely” as noted in table 6. If many of these women were receiving treatment during pregnancy, then per the CDC guideline, the children could not be classified as such as the guideline states “Scenario 4: Congenital Syphilis Unlikely

Any neonate who has a normal physical examination and a serum quantitative nontreponemal serologic titer equal to or less than fourfold of the maternal titer at delivery and both of the following are true:

The mother’s treatment was adequate before pregnancy. The mother’s nontreponemal serologic titer remained low and stable (i.e., serofast) before and during pregnancy and at delivery (e.g., VDRL ≤1:2 or RPR ≤1:4). I am totally open to the possibility that I am misinterpreting this but in general treatment during pregnancy precludes being classified as unlikely. I do not feel that this appreciably changes the results from the paper, but I do think it is worth mentioning. Furthermore, if the

I understand what you mean, but as the CDC Scenario 4 is defined as: "…. a newborn as described in scenario 3, but whose mother was properly treated before 16WG, with stable, low VDRL follow-up” (l. 146). So maybe I am misinterpreting but an appropriate treatment during pregnancy does not prevent from being classified as CDC Scenario 4. My hypothesis is that the word “unlikely” is maybe poorly translated from French to English. I would have preferred “low probability scenario” but the word “unlikely” had been choose by the CDC guidelines. I totally agree with you that the risk of congenital syphilis of a child classified in CDC Scenario 4 is not totally equal to zero. However, the probability is weak, upon to me.

Thank you for mentioning this issue because it troubled me also.

11) Reviewer: « While revising your submission, please upload your figure files to the Preflight Analysis and Conversion Engine (PACE) digital diagnostic tool, https://pacev2.apexcovantage.com/. PACE helps ensure that figures meet PLOS requirements. To use PACE, you must first register as a user. Registration is free. Then, login and navigate to the UPLOAD tab, where you will find detailed instructions on how to use the tool. If you encounter any issues or have any questions when using PACE, please email PLOS at figures@plos.org. Please note that Supporting Information files do not need this step. »

The two figures have been downloaded and converted on PACE, guaranteeing figures meet PLOS requirements. The are uploaded in two different files under the name “fig1.tif” and ‘fig2.tif”.

---

## [Editor Report · Decision Letter 1]

20 Aug 2024

Fetal and neonatal outcomes in syphilis infected pregnant women in Reunion Island : an observationnal retrospective multicentric study.

PONE-D-24-05431R1

Dear Dr. CRAMEZ,

We’re pleased to inform you that your manuscript has been judged scientifically suitable for publication and will be formally accepted for publication once it meets all outstanding technical requirements.

Kind regards,

Everton Falcão de Oliveira, Ph.D

Academic Editor

PLOS ONE
---

## [Editor Report · Acceptance letter]

27 Aug 2024

PONE-D-24-05431R1 

PLOS ONE

Dear Dr. CRAMEZ, 

I'm pleased to inform you that your manuscript has been deemed suitable for publication in PLOS ONE. Congratulations! Your manuscript is now being handed over to our production team.

Kind regards, 

on behalf of

Dr. Everton Falcão de Oliveira 

Academic Editor

PLOS ONE